# Predictors of institutional delivery service utilization among women in Northern region of Ghana

Abdul Gafaru Mohammed[1]*, Ruth Nimota Nukpezah[2], Harriet Bonful[1], Hilarius Paul Asiwome Kosi Abiwu[3], Charles Lwanga Noora[1], Alice Sallar Adams[4], Jennifer Nai-Dowetin[4], Ernest Kenu[1]

1 Department of Epidemiology and Disease Control, University of Ghana, Accra, Ghana, 2 School of Nursing and Midwifery, University for Development Studies, Tamale, Ghana, 3 Northern Regional Health Directorate, Ghana Health Service, Tamale, Ghana, 4 Ghana Field Epidemiology and Laboratory Training Programme, Accra, Ghana

* mohammedabdulgafaru46@gmail.com

## Abstract

### Introduction

An increase in home delivery among expectant mothers may likely lead to high maternal and newborn morbidities and mortalities. Despite the policy on free maternal healthcare in Ghana under the National Health Insurance Scheme (NHIS) since 2007, more than 25% of deliveries still occur outside health facilities in northern Ghana. Use of safe and effective delivery services including place of delivery is an important component of the Safe Motherhood concept. Hence, assessing predictors of institutional delivery could contribute to improving birth outcomes in the Northern Region.

### Methods

We conducted a community-based cross-sectional survey of 310 women aged 15–49 years old who had given a live birth between January 2022 and January 2023, using a simple random sampling approach. Using a semi-structured questionnaire, we collected data on mothers' background characteristics, place of delivery for their most recent birth and reported health facility factors. Descriptive analyses and multiple logistic regression models were performed to identify factors associated with institutional delivery at a 5% significance level.

### Results

Of 310 women in the study, the prevalence of institutional delivery was 79%(245) in their most recent births. More than 60%(200/310) of the women were married and 53%(163/310) had no formal education. Being married (adjusted odds ratio {aOR}=2.8, 95%CI:1.48–5.32), the presence of skilled health personnel at post

**Data availability statement:** All relevant data are within the paper and its Supporting Information files.

**Funding:** This research work was funded by the principal investigator who doubles as the corresponding author. The funder was involved in the design, data collection, data analysis, manuscript drafting, and the decision to publish the work.

**Competing interests:** The authors have declared that no competing interests exist.

(aOR=2.9, 95%CI:1.54–5.43), reported positive attitude of health workers towards their clients (aOR=1.8, 95%CI:1.03–3.23) and positive community perception of health facility delivery (aOR=3.8, 95%CI:1.64–8.71) were associated with increased odds of institutional delivery.

## Conclusions

Our study identified multiple predictors of institutional delivery; marital status, the presence of skilled health personnel at health facilities, the perceived attitude of health workers and community perception. The research team organized discussions on institutional delivery services with community members in five selected districts in the region. We recommend the Ministry of Health should develop well-defined care packages targeting unmarried pregnant women, negative health worker attitudes and negative community perceptions.

## Introduction

More than 500,000 women lose their lives annually during pregnancy and childbirth worldwide, majority (94%) of whom are often from developing countries [1]. Sub-Saharan Africa (SSA) accounts for over 66% of maternal deaths in developing countries [1]. One out of every 26 women in SSA dies from complications related to pregnancy and childbirth [2]. In fact, thirty of the forty countries with the highest maternal death rates globally are from SSA [2].

Home deliveries are still very rampant in developing countries (68.7%), as compared to developed countries (1.3%) [3]. In sub-Saharan Africa, the prevalence of health facility delivery ranges from 23% in Chad to 94% in Gabon, with more than half of the countries recording less than 70% [2].

According to the 2022 Demographic Health Survey (DHS) in Ghana, 86% of live births in the 2 years preceding the survey were delivered in a health facility. This implies more than 10% of children are still delivered outside the health facility setting. The Northern region of Ghana continues to experience one of the lowest institutional delivery rates in the country, with only 70.3% of births occurring in health facilities. Surveys conducted in Brong Ahafo region and Chereponi district of northern Ghana, among 138 and 440 women respectively, revealed health facility deliveries ranged from 38–52% [3,4].

Over the years, the government of Ghana has attempted to improve access to maternal health care services. In 2003, the government introduced the waiver of delivery fees, and by 2005, fees on delivery care were abolished in all the country's regions [5]. This was followed by the introduction of the National Health Insurance Scheme (NHIS) in 2005, which allows all pregnant women under the scheme to have free access to maternal health care services, including antenatal care, delivery services, postnatal care, and neonatal care [6].

To achieve target 3.1 of the third Sustainable Development Goal of ensuring healthy lives and promoting well-being for all at all ages, it is important to ensure all deliveries occur in the health facility setting.

Use of safe and effective delivery services including place of delivery is an important component of the Safe Motherhood concept [1]. Studies on health facility delivery have demonstrated that multiple factors influence the decision to use such a service. Interestingly, while certain factors are significant in determining the use of skilled delivery services in some studies, these same factors were found to be insignificant in others. Individual factors such as maternal age, education, marital status, parity, household factors such as family size, household wealth, and community and environmental factors such as region, community health infrastructure, available health facilities, and distance to health facilities have been identified to operate in diverse contexts to determine the use of institutional delivery services [3,7–9].

Increased deliveries outside the health facility setting among women of reproductive age (WRA) may likely lead to high maternal and newborn morbidities and mortalities as a consequence of complications related to the delivery [10]. Despite the established consequences of deliveries outside a health facility among WRA, there is a dearth of knowledge on the prevalence and predictors of health facility delivery in the Northern region. Also, published studies from the region on institutional delivery service utilisation are mostly descriptive studies or focus on particular districts [11,12] and do not thoroughly investigate the correlates of institutional delivery service utilisation at the regional level. Understanding the factors influencing health facility deliveries in the Northern region is crucial for designing targeted interventions to improve maternal health outcomes. This study aims to identify the prevalence and predictors of institutional delivery service utilization at the regional level, providing evidence-based insights that can guide policymakers in formulating strategies to increase health facility deliveries and ultimately reduce maternal and neonatal mortality in the region. The theoretical framework of the study will be guided by the Health Belief Model (HBM) and elements from Andersen's Behavioral Model of Health Services Use. These models offer a combined approach to understanding how individual beliefs, social context, and access to resources influence institutional delivery. This framework will guide the study in identifying key predictors of service utilization and provide a basis for designing targeted interventions that address specific barriers and motivators identified in the Northern region of Ghana.

## Methods

### Study design

We conducted a community-based cross-sectional study among 310 reproductive-age (15–49 years) women in the Northern region of Ghana. We collected data on participants' background characteristics, choice of place of delivery, and health facility factors using a semi-structured questionnaire. Data was collected from women who had a live birth between January 2022 and January 2023. The data collection period was March 10th – April 2nd 2023. We conducted descriptive analysis and calculated adjusted odds ratios at 95% confidence intervals to identify factors independently associated with health facility delivery.

### Study setting

This research was carried out in the Northern region of Ghana. Among Ghana's 16 regions, the Northern region is one of the most populous, with a projected population of 2,479,461. The region has over 200,000 women of reproductive age [13]. Tamale, the largest city in the region, serves as the capital. The region has 16 administrative health districts. More than two hundred health facilities are licenced to provide health care in the area. These health facilities include hospitals, polyclinics, health centres, Community-based Health Planning and Services (CHPS) and maternity homes. These health facilities provide general medical services including maternal health services to the region's populace and surrounding areas. All pregnant women in the region registered under the NHIS have free access to maternal health care services.

### Study population and eligibility

The study population consisted of women between the ages of 15 and 49 who lived in Ghana's Northern region. All women aged 15–49 years old who had given a live birth between January 2022 and January 2023 and resided in the

region were included in the study. Visitors to the region and those severely ill were excluded from the study. Also, women who met our inclusion criteria but refused to participate for personal or health reasons were excluded from the study.

### Study variables

**Dependent variable.** The dependent variable in the study was the place of delivery in the most recent birth. The variable was binary (Health facility/Home).

**Independent variables.** The independent variables were divided into three groups, sociodemographic characteristics of the participants, health facility-level factors, and community-level factors. The sociodemographic characteristics included occupation, religion, marital status, educational level, income level, husband education, and husband occupation. Health facility-level factors include the availability of a health facility, the attitude of health workers, the presence of health workers at post, and the distance to the nearest health facility. The community-level factors included the availability of TBAs in the community, community perception, the attitude of partners towards health facility delivery, and the availability of transportation to health facilities.

### Sample size determination and sampling process

The sample size for the study was estimated using the Cochrane sample size estimation formula [n = (Z2 pq)/d2]. Using Z = the normal distribution at 95% confidence level which corresponds to 1.96, n = minimum sample size, p = prevalence of home delivery = 28% found in a study conducted in Ghana [14], q = (1- p) and d = precision of 5% = 0.05, we estimated a minimum sample size of 310. A multistage sampling approach was used to sample women of reproductive age (WRA) for the study [15]. Eight districts were randomly selected from the 16 districts in the Northern region. In selecting the districts, the names of all 16 districts in the Northern region were written on pieces of paper and placed in a box. The box was shaken vigorously after which 8 pieces of paper were randomly picked from the box. The 8 names on the papers selected were the sampled districts used for the study. Communities in the selected districts ranged from 20–35. The communities were categorized under stratum A (list of urban communities in the district) and stratum B (list of rural communities in the district). In each of the districts, the names of communities in each stratum were written on pieces of paper and placed in a box, shaken and one piece of paper randomly selected from each stratum. This was repeated for each district until 16 communities were randomly sampled for the study. A probability proportionate to size sampling approach was used to determine the number of women to sample from each of the selected communities, using the projected population of WRA for 2021, obtained from the district health directorates. A systematic random sampling approach was used to select the participants' houses. In each of the selected communities, we started data collection from the town chief or the community leader's house within the community. If the community had no chief or leader, the starting point was an influential person's house (a woman or religious leader). Considering the arrangement of building structures from the community leader's or chief's house, from the starting point, the next house was the K$^{th}$ (household selection interval) house away in the southward or northward direction. For each community, the sampling interval (K) was determined as the total number of households divided by the number of women to be sampled from the community. Households were selected in this direction until there were no more households. Then, households were selected westward and then eastward in a zigzag fashion. Households were visited in this sequence until the required respondents were obtained.

The next community (the same classification as the first (urban or rural) was added within the same district in smaller communities where all respondents could not be obtained.

A respondent was recruited at the starting point and every kth household starting from the southward direction was visited. If two or more women in a household met the inclusion criteria, one respondent was randomly selected by balloting from the list of potential respondents (Table 1).

**Table 1. Districts and the number of women sampled.**

| No | District | Number of women sampled |
|---|---|---|
| 1 | Tamale metropolis | 65 |
| 2 | Tolon district | 42 |
| 3 | Savelugu municipality | 48 |
| 4 | Kumbungu district | 39 |
| 5 | Mion district | 31 |
| 6 | Sagnarigu district | 35 |
| 7 | Tatale district | 37 |
| 8 | Saboba district | 41 |

### Training of research assistants and pretesting

Before data collection, five final-year nursing students of the University for Development Studies were recruited and trained for data collection. A 3-day training exercise was conducted by the principal investigator and a resource person (field epidemiologist) from the Ghana Field Epidemiology and Laboratory Training Programme (GFELTP). Research assistants were taken through kobo-collect data collection methods, how to obtain informed consent, and the COVID-19 preventive measures to observe during the data collection process. The designed data collection tool was pretested in the Greater Accra region using 20 sampled respondents. Mistakes detected at the pretesting stage were addressed before the main data collection.

### Data collection process

Research assistants administered a structured questionnaire during the data collection. They conducted face-to-face interviews with selected eligible participants in their various communities. The semi-structured questionnaire was designed and deployed in the Kobo-collect toolbar for administration. Sociodemographic variables such as age, marital status, ethnic group, religion, education level, and employment history were obtained using the questionnaire. The choice of place of delivery (home/health facility), the availability of health facilities and distance to the nearest health facility and the attitude of health workers were also elicited.

### Data management and statistical analysis

The data was extracted from the Kobo-collect tool in Microsoft Excel format and cleaned. Extracted data with missing information on the place of delivery for their most recent birth were excluded from the analysis. For analysis, data was loaded into STATA software version 16.0. Categorical variables were presented as proportions and frequencies and presented in tables. Cross-tabulations were used to determine prevalence. A logistic regression analysis was employed to establish the degree of the association. The adjusted logistic regression model's variables were selected using a forward stepwise variable selection approach. The adjusted odds ratios and their 95% confidence intervals were presented. At a 5% significance level, a significant correlation was determined. Robust standard errors was used to adjust for clustering in the sampling design with community ID used as the clustering variable.

### Ethical clearance

The Ghana Health Service Ethics Review Committee granted the ethical clearance for the study (GHS-ERC:025/02/23). All participants provided verbal and written informed consent for participation in the study. Consent was obtained from the parents or husbands of women aged less than 18 years, after which they signed an assent form. Data was collected devoid of personal identifiers such as names and contacts. Collected data was accessible by only the principal investigator and the academic supervisor. All preventive measures against COVID-19 were taken to prevent the spread of the disease between study participants and research assistants.

## Results

### Socio-demographic characteristics of study participants, Northern Region

Overall, 340 women were surveyed in the study and 91.2% (310/340) agreed to participate in the study. The average age of the women was 30.9±6.2 years. The majority of women (116; 37.4%) were within the age group 21–30 years and housewives (141; 45.5%). More than two-thirds (263; 84.9%) of the women studied were Muslims, about three-quarters (240; 77.4%) were married and more than half (163; 52.6%) had no formal education (Table 2).

**Table 2. Socio-demographic characteristics of reproductive-aged women, Northern Region, 2023.**

| Variables | Frequency (n) | Percentages (%) |
|---|---|---|
| **Participant's age (years) [mean±sd]** | 30.9±6.2 | |
| < 21 | 27 | 8.7 |
| 21 - 30 | 116 | 37.4 |
| 31 - 40 | 89 | 28.7 |
| ≤ 41 | 78 | 25.2 |
| **Occupation** | | |
| Housewife | 141 | 45.5 |
| Farmer | 14 | 4.5 |
| Trader | 133 | 42.9 |
| State employed | 22 | 7.1 |
| **Religion** | | |
| Christianity | 37 | 11.9 |
| Islam | 263 | 84.9 |
| Traditionalist | 10 | 3.2 |
| **Marital Status** | | |
| Single (Never married) | 39 | 12.6 |
| Divorced | 31 | 10.0 |
| Married | 240 | 77.4 |
| **Education** | | |
| No formal education | 163 | 52.6 |
| Elementary | 96 | 31.0 |
| Secondary | 33 | 10.6 |
| Tertiary | 18 | 5.8 |
| **Husband education** | | |
| No formal education | 110 | 45.8 |
| Elementary | 70 | 29.2 |
| Secondary | 22 | 9.2 |
| Tertiary | 38 | 15.8 |
| **Husband occupation** | | |
| Farmer | 129 | 53.8 |
| State employed | 44 | 18.3 |
| Trader | 67 | 27.9 |
| **Monthly Income (GH₵)** | | |
| 0–100.00 | 209 | 67.4 |
| 101.00–500.00 | 78 | 25.2 |
| >500.00 | 23 | 7.4 |

### Health facility-level factors among reproductive-aged women, Northern Region

The majority (250; 80.6%) of the women reported having a health facility situated in their community. More than two-thirds (239; 77.1%) had to travel for 5–10 km to access a health facility. Most women (251; 81.0%) were registered under the national health insurance scheme. Almost all (296; 95.5%) of the women mentioned the availability of skilled health professionals in the health facilities they visit. On their perceived attitude of the health professionals, the majority (201; 65.0%) stated the health professionals demonstrated a good attitude (Table 3).

### Choice of place of delivery among reproductive-aged women and community level factors, Northern Region

Almost all (302; 97.4%) of the women interviewed reported the presence of traditional birth attendants in their communities. More than 50% (126) of the women said their husbands perceived health facility delivery to be bad. Regarding the community's perception of health facility delivery, most (265; 85.8%) of the women said the community perceives health facility delivery positively. On the place of delivery for their most recent birth, 79.0% (95%CI:74.1–83.4) delivered at a health facility (Table 4).

### Factors associated with the utilization of institutional delivery services among reproductive-aged women, Northern Region

At the multivariate logistic regression analysis level, being married (aOR = 5.54, 95%CI: 3.03–10.14), the presence of skilled health personnel (aOR = 2.65, 95%CI: 1.42–4.94), the positive attitude of health workers towards their clients (aOR = 1.96, 95%CI: 1.08–3.54) and the positive community perception of health facility delivery (aOR = 3.17, 95%CI: 1.34–7.47) were associated with increased odds of delivering in a health facility (Table 5).

**Table 3. Health facility-level factors, Northern Region.**

| Variables | Frequency (n) | Percentages (%) |
|---|---|---|
| **Availability of health facility** | | |
| Unavailable | 60 | 19.4 |
| Available | 250 | 80.6 |
| **Distance to health facility** | | |
| <5km | 54 | 17.4 |
| 5–10 km | 239 | 77.1 |
| >10 km | 17 | 5.5 |
| **National Health Insurance Scheme (NHIS) ownership** | | |
| Yes | 251 | 81.0 |
| No | 59 | 19.0 |
| **Availability of personnel** | | |
| Unavailable | 93 | 30.0 |
| Available | 217 | 70.0 |
| **Attitude of Health workers** | | |
| Poor | 108 | 35.0 |
| Good | 201 | 65.0 |

**Table 4.** Utilization of institutional delivery services and community-level factors, among reproductive-aged women, Northern Region.

| Variables | Frequency (n) | Percentages (%) |
|---|---|---|
| **Availability of Traditional Birth Attendants (TBAs)** | | |
| No | 8 | 2.6 |
| Yes | 302 | 97.4 |
| **Attitudes of the husband towards facility delivery** | | |
| Bad | 126 | 52.5 |
| Good | 114 | 47.5 |
| **Community perception** | | |
| Negative | 44 | 14.2 |
| Positive | 265 | 85.8 |
| **Availability of transport** | | |
| No | 124 | 40.0 |
| Yes | 186 | 60.0 |
| **Place of delivery** | | |
| Home | 65 | 21.0 |
| Health facility | 245 | 79.0 |

**Table 5.** Logistic regression analysis for factors associated with the utilization of institutional delivery services among reproductive-aged women, Northern Region.

| Variables | Place of delivery | | COR (95%CI) | P – value | AOR (95%CI) | P – value |
|---|---|---|---|---|---|---|
| | Home n (%) | Health facility n (%) | | | | |
| **Marital Status** | | | | | | |
| Never married/Divorced | 33 (47.1) | 37 (52.9) | Ref | | Ref | |
| Married | 32 (13.3) | 208 (86.7) | 5.80 (3.18 10.55) | 0.001 | 5.54 (3.03 10.14) | 0.001 |
| **Community perception** | | | | | | |
| Negative | 22 (50.0) | 22 (50.0) | Ref | | Ref | |
| Positive | 43 (16.2) | 222 (83.8) | 1.96 (1.11 3.44) | 0.019 | 3.17 (1.34 7.47) | 0.008 |
| **Attitude of health workers** | | | | | | |
| Poor | 31 (28.7) | 77 (71.3) | Ref | | Ref | |
| Good | 34 (16.9) | 167 (83.1) | 1.97 (1.13 3.45) | 0.016 | 1.96 (1.08 3.54) | 0.025 |
| **Presence of skilled health personnel at health facilities** | | | | | | |
| No | 34 (36.6) | 59 (63.4) | Ref | | Ref | |
| Yes | 31 (14.3) | 186 (85.7) | 3.45 (1.95 6.10) | 0.001 | 2.65 (1.42 4.94) | 0.002 |

## Discussion

The utilization of institutional delivery services by reproductive-age women at the time of delivery is instrumental to the health and well-being of both the mother and the newborn. The government of Ghana has over the years implemented various measures to encourage women in both rural and urban areas to deliver in health facilities and not at home. This study presents a better understanding of the issue of health facility delivery in the Northern region and various factors or indicators that can be targeted to reduce or prevent delivery outside a health facility.

The study revealed that more than 70% of the reproductive-aged women gave birth in a health facility for their most recent birth. The prevalence reported in this study is similar to the findings of the 2017–2018 multiple indicator survey, which estimated institutional delivery in Ghana to be at 73% [16]. The reported prevalence level of health facility delivery in this study is much higher than the prevalence reported by other studies within Africa. In a study conducted in Zala Woreda and Dodota districts, Ethiopia, 77–80% of the women studied reported delivering at home in their most recent delivery [17]. In a study conducted in the Margibi County of Liberia, more than 90% of the women reported that they delivered at home in their most recent birth [18]. Also, a study conducted in Akure, Nigeria, reported that 81.8% of women delivered at home in their most recent delivery [19]. The disparity in the reported prevalence of health facility delivery in these studies compared to our study could be attributed to the study setting and the inclusion criteria used to recruit the study participants. Whereas our study considered women who had given birth in the previous year the other studies recruited women within the reproductive age or expectant mothers as study participants. This implies that the reported prevalence in these studies was the preferred choice of place of delivery by the study participants whereas this study reported the actual prevalence of health facility delivery among the women after it had occurred. The comparatively high prevalence in our current study could also suggest that more women in their reproductive age in Ghana are getting educated on the need to use institutional delivery services and hence are patronizing these services.

The behaviour of health workers towards their clients is instrumental to the success of any healthcare delivery system. Health workers are, by code, required to treat their clients with dignity and respect at all times. When these health workers fail to adhere to their code, clients may seek help for their health issues elsewhere. This study revealed that the attitude of health workers was significantly associated with health facility delivery among reproductive-age women in the region. Women who rate health workers' attitudes as good had almost two-fold increased odds of delivering in the health facility compared to their counterparts. This finding corroborates the findings of studies conducted in Liberia, among reproductive-aged women where the good attitude of health workers was associated with a 99% odds of decreased home delivery among the women [18]. The attitude of health workers towards their clients, especially expectant mothers, must be explored and improved to increase the utilisation of institutional delivery services. This aligns with the HBM, where perceived benefits (e.g., respectful treatment by health workers and availability of skilled staff) increase the likelihood of seeking institutional delivery. Conversely, negative attitudes of health workers act as barriers, reducing the perceived benefit of facility-based deliveries. Management of health facilities should organise workshops or seminars and educate health workers on human relationships and good behaviours. Also, health nursing institutions such as nursing and medical training institutions should place more emphasis on behavioural science courses where current students will be educated on how to relate to and treat their clients in all situations.

The availability of health workers at various health facilities is instrumental to the utilisation of services offered by the facility. Women who mentioned the availability of skilled health workers at the health facilities in their communities were more inclined to deliver at a health facility compared to those who mentioned the constant unavailability of health workers at the health facilities. This finding is consistent with a similar study conducted in the Oti Region, Krachi Nchumuru District of Ghana, where women mentioned the unavailability of health workers as one of the primary factors influencing home delivery in the district [20]. In a similar study conducted in rural Zambia, the shortage of health workers in rural communities caused women to deliver at home [21]. Another qualitative analysis of home delivery in rural Zambia revealed that women neglect facility delivery due to the unavailability of health workers in health facilities [22]. Receiving care from trained health professionals is one of the reasons why expectant mothers deliver at health facilities, with this singular reason, the unavailability of health workers at post will definitely deter women from delivering in the health facility. To increase the utilisation of institutional delivery services, the Government of Ghana must ensure health workers are posted to all health facilities in the country. Also, health facilities' management should ensure that health workers under their watch stay at their posts and provide all the necessary services to their clients.

The perception of community members on health facility delivery services was another factor reported to be associated with institutional delivery service utilization among the study participants. In communities where health facility delivery is perceived to be good, more than three times more women are likely to deliver in a health facility compared to communities where health facility delivery services are negatively perceived. This is consistent with recent studies in Tanzania and Ghana which reported increased health facility delivery to be associated with community norms and perceptions [23,24]. To increase health facility delivery service utilization it is important to demystify the negative perceptions held by members of the community regarding health facility deliveries. In recent times, various pregnancy schools in Ghana have contributed to health facility delivery by educating pregnant women, their partners and other family members on the importance of health facility delivery. Examples of these schools include; Altar of Grace Baby Care School, Mothers Pride Academy, and 31st December Women Movement School among others [25]. These pregnancy schools should consider expanding their services to the rural communities in the northern region of Ghana which will help in making more gains in increasing health facility delivery service utilization.

The findings from the study align with the theoretical framework, supporting the idea that health beliefs (under HBM) and enabling resources (under Andersen's model) significantly affect institutional delivery service utilization. The study's emphasis on improving health workers' attitudes and addressing service availability directly targets the barriers identified by these models, which can help increase institutional delivery rates by enhancing perceived benefits, reducing barriers, and improving enabling factors.

The study was not devoid of limitations; the quantitative nature of the study did not allow for the authors to explore the perception of women on the use of institutional delivery services. Also, data was not collected on COVID-19-related factors that could influence institutional delivery services. Also, the use of only the prevalence of institutional delivery estimates in calculating the study sample size without taking into account any of the predictors of institutional deliveries was another limitation of our study. Further research should be conducted to assess the factors influencing institutional delivery service or home delivery using qualitative approaches in the region.

## Conclusions

Our study identified multiple predictors of health facility delivery: marital status, presence of skilled health personnel at health facilities, the attitude of health workers and community perception. To increase health facility delivery, we recommend the Ministry of Health should institute a policy reform with a well-defined care package targeting unmarried pregnant women and health workers with a negative attitude and community perception. Also, the government should provide incentives for health workers working in rural communities.

## Supporting information

**S1 Data. Datafile.**
(XLSX)

## Acknowledgments

We are grateful to all data collectors and research participants.

## Author contributions

**Conceptualization:** Abdul Gafaru Mohammed, Ruth Nimota Nukpezah, Hilarius Paul Asiwome Kosi Abiwu, Charles Lwanga Noora, Alice Sallar Adams, Jennifer Nai-Dowetin, Ernest Kenu.

**Data curation:** Abdul Gafaru Mohammed, Ruth Nimota Nukpezah, Harriet Bonful, Hilarius Paul Asiwome Kosi Abiwu, Charles Lwanga Noora, Alice Sallar Adams, Jennifer Nai-Dowetin, Ernest Kenu.

**Formal analysis:** Abdul Gafaru Mohammed, Harriet Bonful, Hilarius Paul Asiwome Kosi Abiwu, Charles Lwanga Noora, Alice Sallar Adams, Jennifer Nai-Dowetin, Ernest Kenu.

**Funding acquisition:** Abdul Gafaru Mohammed.

**Investigation:** Abdul Gafaru Mohammed, Hilarius Paul Asiwome Kosi Abiwu, Ernest Kenu.

**Methodology:** Abdul Gafaru Mohammed.

**Project administration:** Abdul Gafaru Mohammed.

**Resources:** Abdul Gafaru Mohammed.

**Software:** Abdul Gafaru Mohammed.

**Supervision:** Abdul Gafaru Mohammed, Ruth Nimota Nukpezah.

**Validation:** Abdul Gafaru Mohammed, Harriet Bonful.

**Visualization:** Abdul Gafaru Mohammed.

**Writing – original draft:** Abdul Gafaru Mohammed, Ruth Nimota Nukpezah, Harriet Bonful, Hilarius Paul Asiwome Kosi Abiwu, Charles Lwanga Noora, Alice Sallar Adams, Jennifer Nai-Dowetin, Ernest Kenu.

**Writing – review & editing:** Abdul Gafaru Mohammed, Ruth Nimota Nukpezah, Harriet Bonful, Hilarius Paul Asiwome Kosi Abiwu, Charles Lwanga Noora, Alice Sallar Adams, Jennifer Nai-Dowetin, Ernest Kenu.

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
