## [Decision Letter · Decision Letter 0]

18 Jul 2023

PONE-D-23-14896Predictors of Institutional Delivery Service Utilization among Women in Northern Region of GhanaPLOS ONE

Dear Dr. Mohammed,

Thank you for submitting your manuscript to PLOS ONE. After careful consideration, we feel that it has merit but does not fully meet PLOS ONE’s publication criteria as it currently stands. Therefore, we invite you to submit a revised version of the manuscript that addresses the points raised during the review process.

Reviewer 1 comments:Methods:

Page 7 : Sample size determination and sampling process

line 146- 150 ; details on how the 8 districts and two communities were randomly selected should be given

How was the sampling frame of 5 arrived at?

How was the southward direction decided to be followed first until there were no more houses before other directions from the start point?

Discussion: Page 16

Line: 267-271; “ The disparity in the reported prevalence…………….” Explain why recruiting women who have given birth in the previous year would give such different prevalence from the women recruited from the reproductive age or expectant mothers?

Any recent studies from the respective countries referenced in your study? Reviewer 2 comments

**PONE-D-23-14896 MANUSCRIPT REVIEWERS COMMENTS**

ABSTRACT

Stop first sentence at” morbidity and mortality”

There on free maternal health care but NHIS

Why did you not study the prevalence and predictors of home delivery? Some reasons why they delivered at home may still remain unknown.

Conclusion: What is “availability of skilled personnel”?

LINES 25-39: The comments above also apply to this section

LINES 47-48 When does a woman become A SINGLE MOTHER? Is it when she is pregnancy or after delivery.? Are they unmarried pregnant women? Teenagers? or their husbands travelled while they were pregnant?

LINES 54-73: These definitions are not necessary in the manuscript

LINE 76: ..in the “reproductive process”….this is not clear. You need to it up from the reference 1.

METHODS

LINES 122: Data collected from 10^th^ March- 2^nd^ April 2023 and compare to -lines 31: Jan 2022 to Jan 2023. Put the two sentences together so that it becomes clear what you want to do.

What is community chairman? Chairman of what?

LINE 163: Needs clarity. Respondents were recruited at the starting point and every fifth house was visited.

LINE 172: What about COVID 19? When you are collecting data in March- April 2023.

RESULTS IN THE TABLES

TABLES 1: Differences between divorced and single. Almost 38% (119) were single women in those rural communities yet 245 husbands had positive or negative attitudes as shown in table 4.

That makes total number of husbands 54 more than your study of 310 respondents. Are some of the women having more than a husband or what. In the same table 4, under husband’s occupation total is 320 also more than your study of 310 respondents.

Where from these errors? How do these errors affect your findings and conclusions?

DISCUSSION

LINES 317-344

How is the impact of pregnancy schools relevant in this study which was in a rural population?. Something you found in your study attracts them to health facilities so discuss that more.

CONCLUSION LINES 329-331: The major issue with the conclusion is that there is no clear understanding of who is a single mother.

The data from tables 1 & 4 are conflicting on marital status of the women, attitudes of the husbands and number of husbands in the various occupations listed.

REFEENCES

All references must conform to Journals referencing style/format

Some references are incomplete eg 22 and 23 are not complete

Reviewer 3 comments

Introduction

Use of safe and effective delivery services including place of delivery is an important component of the Safe Motherhood concept. Hence, assessing predictors of institutional delivery could contribute to improving birth outcomes in the study setting.

Minor revisions

1. There are more recent estimates of global maternal mortality. The authors are advised to use these (eg Trends in maternal mortality 2000 to 2020: estimates by WHO, UNICEF, UNFPA, World Bank Group and UNDESA/Population Division. Geneva: WHO 2023).

2. Line 59: The definition given is that of maternal mortality ratio rather the rate.

3. A few sentences need to be edited eg line 95 “…sustainable development goal…” should be written as “…Sustainable Development Goal...”. Lines 139 and 226 should be in the past tense; ie reside and perceive respectively should be in the past tense.

4. Some acronyms were used without defining them at the first instance eg CHPS, DHS, MIS, WHO etc. Acronyms/abbreviations used in Tables should all be defined below the table.

5. Study setting: The authors should state the population of WRA in the region.

6. Methods: The selected districts together with the number of women selected from each district should be stated.

7. Line 188: Although the authors stated that there were some exclusions due to missing data, there is no evidence to support this as the estimated sample size (310) is exactly the same as the number of women surveyed. It appears all women approached agreed and participated in the study. Otherwise, they should state the number of women who were excluded.

8. Ethical considerations: The authors should clearly describe how consent was obtained from minors (ie women <18 years of age)

9. Provide refs for the sampling procedure described on page 8 and pregnancy schools in Ghana (lines 317-324), which could be of interest to readers.

10. The authors should provide the mean age, standard deviation and the range.

11. In Table 4, there is no need to indicate ** against significant p-values as the authors have indicated in the methods section that p<0.05 will be considered statistically significant. I believe all stated p-values in the table will be interpreted in that context. The authors rounded off some ORs from Table 4 in the text (see lines 237-247). The authors should state ORs in the text as they are in the table for ease of reference. They also repeated some ORs with their 95% CIs within the same paragraph and in some instances the ORs were different (eg lines 239 and 244; and 237 and 246). As much as possible, the authors should avoid repeating results especially within the same paragraph. They can make their point without repeating the results.

12. The authors should provide areas for further research in the conclusion.

Major revisions

1. My major concern with the study is the sample size and its estimation. I do not think a sample of 310 is representative enough of the entire population of women of reproductive age (WRA) in the Northern region of Ghana. What is the population of WRA in the region? The stated sample size only estimated a single proportion of home deliveries without incorporating the predictors of home deliveries (such as the proportions and measures of association eg ORs as used in this study). Besides, no adjustments were made for the community-based sampling technique eg could the design effect be modified?

2. In line 193 the authors stated the criteria for inclusion into the multivariable model as p<0.05 in the univariable analysis. Yet in Table 4, several variables with p>0.05 in the univariable analysis were included in the multivariable analysis (age group, religion, educational level, occupation, husband occupation). The multivariable analysis should be re-run without these covariates ie ensuring that only covariates which meet the inclusion criteria are included in the multivariable model.

3. The authors should discuss the limitations of the study.

We look forward to receiving your revised manuscript.

Kind regards,

Kwaku Asah-Opoku

Academic Editor

PLOS ONE

Reviewers' comments:

Reviewer's Responses to Questions

**Comments to the Author**

1. Is the manuscript technically sound, and do the data support the conclusions?

Reviewer #1: Yes

Reviewer #2: No

Reviewer #3: Yes

2. Has the statistical analysis been performed appropriately and rigorously? 

Reviewer #1: Yes

Reviewer #2: No

Reviewer #3: No

3. Have the authors made all data underlying the findings in their manuscript fully available?

Reviewer #1: No

Reviewer #2: No

Reviewer #3: Yes

4. Is the manuscript presented in an intelligible fashion and written in standard English?

Reviewer #1: Yes

Reviewer #2: Yes

Reviewer #3: Yes

5. Review Comments to the Author

Reviewer #1: Methods:

Page 7 : Sample size determination and sampling process

line 146- 150 ; details on how the 8 districts and two communities were randomly selected should be given

How was the sampling frame of 5 arrived at?

How was the southward direction decided to be followed first until there were no more houses before other directions from the start point?

Discussion: Page 16

Line: 267-271; “ The disparity in the reported prevalence…………….” Explain why recruiting women who have given birth in the previous year would give such different prevalence from the women recruited from the reproductive age or expectant mothers?

Any recent studies from the respective countries referenced in your study?

Reviewer #2: ABSTRACT

Stop first sentence at” morbidity and mortality”

There on free maternal health care but NHIS

Why did you not study the prevalence and predictors of home delivery? Some reasons why they delivered at home may still remind unknown.

Conclusion: What is “availability of skilled personnel”?

LINES 25-39: The comments above also apply to this section

LINES 47-48 When does a woman become A SINGLE MOTHER? Is it when she is pregnancy or after delivery.? Are they unmarried pregnant women? Teenagers? or their husbands travelled while they were pregnant?

LINES 54-73: These definitions are not necessary in the manuscript

LINE 76: ..in the “reproductive process”….this is not clear. You need to it up from the reference 1.

METHODS

LINES 122: Data collected from 10th March- 2nd April 2023 and compare to -lines 31: Jan 2022 to Jan 2023. Put the two sentences together so that it becomes clear what you want to do.

What is community chairman? Chairman of what?

LINE 163: Needs clarity. Respondents were recruited at the starting point and every fifth house was visited.

LINE 172: What about COVID 19? When you are collecting data in March- April 2023.

RESULTS IN THE TABLES

TABLES 1: Differences between divorced and single. Almost 38% (119) were single women in those rural communities yet 245 husbands had positive or negative attitudes as shown in table 4.

That makes total number of husbands 54 more than your study of 310 respondents. Are some of the women having more than a husband or what. In the same table 4, under husband’s occupation total is 320 also more than your study of 310 respondents.

Where from these errors? How do these errors affect your findings and conclusions?

DISCUSSION

LINES 317-344

How is the impact of pregnancy schools relevant in this study which was in a rural population?. Something you found in your study attracts them to health facilities so discuss that more.

CONCLUSION LINES 329-331: The major is

Reviewer #3: Introduction

Use of safe and effective delivery services including place of delivery is an important component of the Safe Motherhood concept. Hence, assessing predictors of institutional delivery could contribute to improving birth outcomes in the study setting.

Minor revisions

1. There are more recent estimates of global maternal mortality. The authors are advised to use these (eg Trends in maternal mortality 2000 to 2020: estimates by WHO, UNICEF, UNFPA, World Bank Group and UNDESA/Population Division. Geneva: WHO 2023).

2. Line 59: The definition given is that of maternal mortality ratio rather the rate.

3. A few sentences need to be edited eg line 95 “…sustainable development goal…” should be written as “…Sustainable Development Goal...”. Lines 139 and 226 should be in the past tense; ie reside and perceive respectively should be in the past tense.

4. Some acronyms were used without defining them at the first instance eg CHPS, DHS, MIS, WHO etc. Acronyms/abbreviations used in Tables should all be defined below the table.

5. Study setting: The authors should state the population of WRA in the region.

6. Methods: The selected districts together with the number of women selected from each district should be stated.

7. Line 188: Although the authors stated that there were some exclusions due to missing data, there is no evidence to support this as the estimated sample size (310) is exactly the same as the number of women surveyed. It appears all women approached agreed and participated in the study. Otherwise, they should state the number of women who were excluded.

8. Ethical considerations: The authors should clearly describe how consent was obtained from minors (ie women <18 years of age)

9. Provide refs for the sampling procedure described on page 8 and pregnancy schools in Ghana (lines 317-324), which could be of interest to readers.

10. The authors should provide the mean age, standard deviation and the range.

11. In Table 4, there is no need to indicate ** against significant p-values as the authors have indicated in the methods section that p<0.05 will be considered statistically significant. I believe all stated p-values in the table will be interpreted in that context. The authors rounded off some ORs from Table 4 in the text (see lines 237-247). The authors should state ORs in the text as they are in the table for ease of reference. They also repeated some ORs with their 95% CIs within the same paragraph and in some instances the ORs were different (eg lines 239 and 244; and 237 and 246). As much as possible, the authors should avoid repeating results especially within the same paragraph. They can make their point without repeating the results.

12. The authors should provide areas for further research in the conclusion.

Major revisions

1. My major concern with the study is the sample size and its estimation. I do not think a sample of 310 is representative enough of the entire population of women of reproductive age (WRA) in the Northern region of Ghana. What is the population of WRA in the region? The stated sample size only estimated a single proportion of home deliveries without incorporating the predictors of home deliveries (such as the proportions and measures of association eg ORs as used in this study). Besides, no adjustments were made for the community-based sampling technique eg could the design effect be modified?

2. In line 193 the authors stated the criteria for inclusion into the multivariable model as p<0.05 in the univariable analysis. Yet in Table 4, several variables with p>0.05 in the univariable analysis were included in the multivariable analysis (age group, religion, educational level, occupation, husband occupation). The multivariable analysis should be re-run without these covariates ie ensuring that only covariates which meet the inclusion criteria are included in the multivariable model.

3. The authors should discuss the limitations of the study.

6. PLOS authors have the option to publish the peer review history of their article (what does this mean? ). If published, this will include your full peer review and any attached files.

**Do you want your identity to be public for this peer review?** For information about this choice, including consent withdrawal, please see our Privacy Policy .

Reviewer #1: **Yes: ** Kareem Mumuni

Reviewer #2: No

Reviewer #3: No

---

## [Author Response · Author response to Decision Letter 0]

24 Jul 2023

Dear Editor and reviewers,

We appreciate all of the valuable comments from the reviewers of our work. We have revised our manuscript according to the reviewers’ comments, questions, and suggestions. We believe that the manuscript has been further improved.

Attached below are detailed responses to all the reviewer’s comments. The responses are shown in green and italicized. Please let us know if you still have any questions or concerns about the manuscript. We will be happy to address them, now promptly

Reviewer 1 comments:

Methods:

Page 7: Sample size determination and sampling process line 146- 150; details on how the 8 districts and two communities were randomly selected should be given. How was the sampling frame of 5 arrived at?

Response

The details on how the sampling of the districts and regions has been included “In selecting the districts, the names of all 16 districts in the Northern region were written on pieces of paper and placed in a box. The box was shaken vigorously after which 8 pieces of paper were randomly picked from the box. The 8 names on the papers selected were the sampled districts used for the study. Communities in the selected districts ranged from 12 – 20. The communities were categorized under stratum A (list of urban communities in the district) and stratum B (list of rural communities in the district). In each of the districts, the names of communities in each stratum were written on pieces of paper and placed in a box, shaken and one piece of paper randomly selected from each stratum. This was repeated for each district until 16 communities were randomly sampled for the study” Line 187 - 195

How was the southward direction decided to be followed first until there were no more houses before other directions from the start point?

Response

Determining the direction to initiate sampling has been included “Considering the arrangement of building structures from the community leader’s or chief’s house, from the starting point, the next house was the Kth (house selection interval) house away in the southward or northward direction. For each community, the sampling interval (K) was determined as the total number of houses divided by the number of women to be sampled from the community. Houses were selected in this direction” Lines 215 – 220

Discussion: Page 16 Line: 267-271; “ The disparity in the reported prevalence…………….” Explain why recruiting women who have given birth in the previous year would give such different prevalence from the women recruited from the reproductive age or expectant mothers?

Response

The reason for the disparity in prevalence based on the participants recruited has been addressed “Whereas our study considered women who had given birth in the previous year the other studies recruited women within the reproductive age or expectant mothers as study participants. This implies that the reported prevalence in these studies was the preferred choice of place of delivery by the study participants whereas this study reported the actual prevalence of health facility delivery among the women after it had occurred” lines 573 – 575

Any recent studies from the respective countries referenced in your study?

Response

Yes, a finding from the 2017 – 2018 multiple indicator survey has been referenced in the study. line 563 - 565

Reviewer 2 comments

ABSTRACT

Stop first sentence at” morbidity and mortality”

Response

The sentence has been revised to effect the change see lines 29 – 30 “An increase in home delivery among expectant mothers may likely lead to high maternal and newborn morbidities and mortalities”

There on free maternal health care but NHIS

Response

The sentence has been addressed to include the NHIS, kindly see lines 30 - 31

Why did you not study the prevalence and predictors of home delivery? Some reasons why they delivered at home may still remain unknown.

Response

This has been stated as an area for further research to be conducted. kindly see lines 660 – 661 “Further research should be conducted to assess the factors influencing institutional delivery service or home delivery using qualitative approaches in the region”

Conclusion: What is “availability of skilled personnel”?

Response

The availability of skilled personnel has been revised to the presence of skilled health personnel at health facilities. Kindly see lines 49 - 50

LINES 25-39: The comments above also apply to this section

LINES 47-48 When does a woman become A SINGLE MOTHER? Is it when she is pregnancy or after delivery.? Are they unmarried pregnant women? Teenagers? or their husbands travelled while they were pregnant?

Response

The single mother as used in the study has been revised to unmarried pregnant women. kindly see line 64 – 66 “We recommend the Ministry of Health should develop well-defined care packages targeting unmarried pregnant women, negative health worker attitudes and negative community perceptions”

LINES 54-73: These definitions are not necessary in the manuscript

Response

The definitions of some indicators as used in this study has been deleted per the review comment

LINE 76: ..in the “reproductive process”….this is not clear. You need to it up from the reference 1.

Response

The phrase reproductive process as used in the background has been revised to “during child birth” kindly see line 71

METHODS

LINES 122: Data collected from 10th March- 2nd April 2023 and compare to -lines 31: Jan 2022 to Jan 2023. Put the two sentences together so that it becomes clear what you want to do.

Response

The two sentences have been kept together to provide more clarity as stated by the reviewer. Kindly see lines 153 – 154 “Data was collected from women who had a live birth between January 2022 and January 2023. The data collection period was March 10th – April 2nd 2023”

What is community chairman? Chairman of what?

Response

The word community chairman as used in the study has been revised to community leader. Kindly see line 214

LINE 163: Needs clarity. Respondents were recruited at the starting point and every fifth house was visited.

Response

How the participants were recruited has been revised to give more clarity to the work. kindly see line 215 – 222 “Considering the arrangement of building structures from the community leader’s or chief’s house, from the starting point, the next house was the Kth (house selection interval) house away in the southward or northward direction. For each community, the sampling interval (K) was determined as the total number of houses divided by the number of women to be sampled from the community. Houses were selected in this direction until there were no more houses in that direction, then houses were selected westward, then eastward in a zigzag fashion. Houses were visited in this sequence until the required respondents were obtained”

LINE 172: What about COVID 19? When you are collecting data in March- April 2023.

Response

The lack of COVID-19 data in the study has been explained as a limitation in the study. kindly see line 658 - 659

RESULTS IN THE TABLES

TABLES 1: Differences between divorced and single. Almost 38% (119) were single women in those rural communities yet 245 husbands had positive or negative attitudes as shown in table 4.

That makes total number of husbands 54 more than your study of 310 respondents. Are some of the women having more than a husband or what. In the same table 4, under husband’s occupation total is 320 also more than your study of 310 respondents.

Where from these errors? How do these errors affect your findings and conclusions?

Response

The differences in the single status and divorced as used in this study have been rectified. Kindly see the revision in line 303, Table 2. Also, the inconsistency in the reported numbers or frequencies has been revised. Table 2 and 4 has been revised to ensure consistency.

DISCUSSION

LINES 317-344

How is the impact of pregnancy schools relevant in this study which was in a rural population?. Something you found in your study attracts them to health facilities so discuss that more.

Response

As recommended, more details on factors such as community perception has been discussed and the statement on the impact of pregnancy schools in the study revised. Kindly see lines 612 - 656

CONCLUSION LINES 329-331: The major issue with the conclusion is that there is no clear understanding of who is a single mother.

Response

The single mother status as used in the conclusion has been revised to unmarried pregnant women. kindly see line 666

REFEENCES

All references must conform to Journals referencing style/format

Some references are incomplete eg 22 and 23 are not complete

Response

References have been revised to ensure consistency. Also, reference 22 and 23 has been completed. Kindly see lines 754 - 763

Reviewer 3 comments

Introduction

Use of safe and effective delivery services including place of delivery is an important component of the Safe Motherhood concept. Hence, assessing predictors of institutional delivery could contribute to improving birth outcomes in the study setting.

Response

The sentence provided by the reviewer has been incorporated into the introduction section of the manuscript abstract to provide more clarity. Kindly see lines 32 – 35

Minor revisions

1. There are more recent estimates of global maternal mortality. The authors are advised to use these (eg Trends in maternal mortality 2000 to 2020: estimates by WHO, UNICEF, UNFPA, World Bank Group and UNDESA/Population Division. Geneva: WHO 2023).

Response

The statement on maternal mortality related to pregnancy and childbirth has been revised to reflect the suggestions made by the reviewer. Kindly see 71

2. Line 59: The definition given is that of maternal mortality ratio rather the rate.

Response

The definitions of key terms used in the study have been deleted from the manuscript as suggested by other reviewers

3. A few sentences need to be edited eg line 95 “…sustainable development goal…” should be written as “…Sustainable Development Goal...”. Lines 139 and 226 should be in the past tense; ie reside and perceive respectively should be in the past tense.

Response

The revisions suggested have been affected. Kindly see line 124 (Sustainable Development Goal) and line 177 (resided) and line 373 (perceived)

4. Some acronyms were used without defining them at the first instance eg CHPS, DHS, MIS, WHO etc. Acronyms/abbreviations used in Tables should all be defined below the table.

Response

All acronyms used in the manuscript have been defined at its first usage. Kindly see the revision line 163 (Community-based Health Planning and Services - CHPS)

5. Study setting: The authors should state the population of WRA in the region.

Response

The population of women of reproductive age in the study area has been stated. Kindly see line 161

6. Methods: The selected districts together with the number of women selected from each district should be stated.

Response

The selected districts and the number of women selected from each district has been presented in Table 1, kindly see line 228

7. Line 188: Although the authors stated that there were some exclusions due to missing data, there is no evidence to support this as the estimated sample size (310) is exactly the same as the number of women surveyed. It appears all women approached agreed and participated in the study. Otherwise, they should state the number of women who were excluded.

Response

Information on the number of study participants who were approached and those who accepted to participate in the study has been stated by the authors. Kindly see lines 297 – 298 “Overall, 340 women were surveyed in the study and 91.2% (310/340) agreed to participate in the study”

8. Ethical considerations: The authors should clearly describe how consent was obtained from minors (ie women <18 years of age)

Response

How consent was obtained from minors in the study has been clearly stated in the ethical consideration section of the manuscript. Kindly see lines 279 – 280 “. Consent was obtained from the parents or husbands of women aged less than 18 years. Also, an assent form was obtained from the minors”

9. Provide refs for the sampling procedure described on page 8 and pregnancy schools in Ghana (lines 317-324), which could be of interest to readers.

Response

References have been provided for the sampling procedure, kindly see line 186. A reference has also been provided for the pregnancy schools stated in the manuscript. Kindly see 624.

10. The authors should provide the mean age, standard deviation and the range.

Response

The variable age was collected as a categorical variable, this makes it impossible to present the summary statistics as proposed by the reviewer

11. In Table 4, there is no need to indicate ** against significant p-values as the authors have indicated in the methods section that p<0.05 will be considered statistically significant. I believe all stated p-values in the table will be interpreted in that context. The authors rounded off some ORs from Table 4 in the text (see lines 237-247). The authors should state ORs in the text as they are in the table for ease of reference. They also repeated some ORs with their 95% CIs within the same paragraph and in some instances the ORs were different (eg lines 239 and 244; and 237 and 246). As much as possible, the authors should avoid repeating results especially within the same paragraph. They can make their point without repeating the results.

Response

Table 4 and its text interpretation has been revised to ensure consistency in the results reported. Always the repetition of ORs has been revised. The ** used in the table to indicate significant p values has been deleted. Kindly see lines 396 - 401

12. The authors should provide areas for further research in the conclusion.

Response

Areas of further research have been provided as suggested by the reviewers. Kindly see lines 660 – 661 “Further research should be conducted to assess the factors influencing institutional delivery service or home delivery using qualitative approaches in the region”

Major revisions

1. My major concern with the study is the sample size and its estimation. I do not think a sample of 310 is representative enough of the entire population of women of reproductive age (WRA) in the Northern region of Ghana. What is the population of WRA in the region? The stated sample size only estimated a single proportion of home deliveries without incorporating the predictors of home deliveries (such as the proportions and measures of association eg ORs as used in this study). Besides, no adjustments were made for the community-based sampling technique eg could the design effect be modified?

Response

The sample size was based on the number of deliveries recorded since our target population was women who had given birth a year prior to the study. So, although there is a large population of WRA in the region, the estimated number of deliveries in the region is much lower. Also, the design effect was adjusted for in the data analysis, where robust standard errors using the type of community as a clustering variable was conducted.

2. In line 193 the authors stated the criteria for inclusion into the multivariable model as p<0.05 in the univariable analysis. Yet in Table 4, several variables with p>0.05 in the univariable analysis were included in the multivariable analysis (age group, religion, educational level, occupation, husband occupation). The multivariable analysis should be re-run without these covariates ie ensuring that only covariates which meet the inclusion criteria are included in the multivariable model.

Response

The strategy used in selecting variables for the adjusted logistic regression analysis has been revised. Kindly see lines 270 – 271 “The adjusted logistic regression model's variables were selected using the stepwise regression approach”

3. The authors should discuss the limitations of the study.

Response

The limitations of the study have been discussed as suggested by the reviewer. Kindly see lines 657 – 661 “The study was not devoid of limitations; the quantitative nature of the study did not allow for the authors to explore the perception of women on

---

## [Decision Letter · Decision Letter 1]

21 Aug 2023

PONE-D-23-14896R1Predictors of Institutional Delivery Service Utilization among Women in Northern Region of GhanaPLOS ONE

Dear Dr. Mohammed,

Thank you for submitting your manuscript to PLOS ONE. After careful consideration, we feel that it has merit but does not fully meet PLOS ONE’s publication criteria as it currently stands. Therefore, we invite you to submit a revised version of the manuscript that addresses the points raised during the review process.

We look forward to receiving your revised manuscript.

Kind regards,

Kwaku Asah-Opoku

Academic Editor

PLOS ONE

Reviewers' comments:

Reviewer's Responses to Questions

**Comments to the Author**

1. If the authors have adequately addressed your comments raised in a previous round of review and you feel that this manuscript is now acceptable for publication, you may indicate that here to bypass the “Comments to the Author” section, enter your conflict of interest statement in the “Confidential to Editor” section, and submit your "Accept" recommendation.

Reviewer #1: All comments have been addressed

Reviewer #2: All comments have been addressed

Reviewer #3: (No Response)

2. Is the manuscript technically sound, and do the data support the conclusions?

Reviewer #1: (No Response)

Reviewer #2: Yes

Reviewer #3: Partly

3. Has the statistical analysis been performed appropriately and rigorously? 

Reviewer #1: (No Response)

Reviewer #2: Yes

Reviewer #3: No

4. Have the authors made all data underlying the findings in their manuscript fully available?

Reviewer #1: (No Response)

Reviewer #2: Yes

Reviewer #3: (No Response)

5. Is the manuscript presented in an intelligible fashion and written in standard English?

Reviewer #1: (No Response)

Reviewer #2: Yes

Reviewer #3: Yes

6. Review Comments to the Author

Reviewer #1: (No Response)

Reviewer #2: No more comments

The authors have answered all the questions that were asked in the initial review.

Recommended for publication

Reviewer #3: Thank you very much for carrying out most of the suggested revisions. There are still a few revisions that need to be addressed to further improve the quality of the manuscript.

1. My “introduction” was meant to be a preamble to my review and not meant to be incorporated into the abstract. Hence it was not part of the suggested revisions. However, if the authors find it useful to include it in the introduction of the abstract, that is fine with me. While the authors have included the statements (from my introduction) in the abstract within the main manuscript, the statements not in the abstract that is in the preliminary pages of the manuscript.

2. In response to suggested minor revision # 7 the authors wrote: Kindly see lines 297 – 298 “Overall, 340 women were surveyed in the study and 91.2% (310/340) agreed to participate in the study”.

The authors should kindly state at least some of the reasons why people were excluded.

3. In response to suggested minor revision #10, the authors wrote: “The variable age was collected as a categorical variable, this makes it impossible to present the summary statistics as proposed by the reviewer.”

It is important to state the mean and standard deviation for age as a continuous variable. Age as a continuous variable should not have been categorized/grouped at the data collection stage. The authors should have collected individual ages and grouping/categorization done at the analysis stage. If this was not done, there are still ways of estimating the mean and standard deviation of the data such as using the midpoint of each age group and the respective frequencies. The authors should kindly consult a statistician on how to do this.

4. In response to suggested major revision #1, the authors stated that:

The sample size was based on the number of deliveries recorded since our target population was women who had given birth a year prior to the study. So, although there is a large population of WRA in the region, the estimated number of deliveries in the region is much lower. Also, the design effect was adjusted for in the data analysis, where robust standard errors using the type of community as a clustering variable was conducted.

I think my fundamental point is that a sample estimation for predictors of institutional deliveries that makes use of only the prevalence of institutional deliveries without taking into account any of the predictors of institutional deliveries is problematic and may not be adequately powered to detect these predictors. The use of robust standard errors will not adequately deal with the issue of community-based sampling. Given that the data has already been collected, one way round this is to discuss the inadequately powered sample size as a limitation of the study.

5. In response to suggested major revision #2, the authors stated that:

The strategy used in selecting variables for the adjusted logistic regression analysis has been revised. Kindly see lines 270 – 271 “The adjusted logistic regression model's variables were selected using the stepwise regression approach”.

Which technique was used in adding or eliminating the variables in the stepwise approach? In the two common variable selection approaches in logistic regression (forward selection and backward elimination (selection) methods), the final multivariable model includes variables that are significantly associated with the outcome of interest (based on the authors’ criteria, here p<0.05, using a model fitness test). This will typically exclude all variables that were not significant in the univariable analysis (unless they were considered to be a priori). In addition, it may exclude some variables which were significant in the univariable analysis but were not significantly associated with outcome of interest in the multivariable model based on the model fitness test. Hence, the following variables which were not significant in the univariable model (all p>0.05) should NOT be included in the final multivariable model: age group, religion, educational level, occupation, husband occupation, National Health Insurance Scheme ownership, and attitudes of husbands towards facility delivery.

The authors can please refer to:

Hosmer Jr DW, Lemeshow S and Sturdivant RX (2013). Applied Logistic Regression. 3rd Edition. New York: John Wiley and Sons, Inc.

7. PLOS authors have the option to publish the peer review history of their article (what does this mean? ). If published, this will include your full peer review and any attached files.

**Do you want your identity to be public for this peer review?** For information about this choice, including consent withdrawal, please see our Privacy Policy .

Reviewer #1: **Yes: ** Kareem Mumuni

Reviewer #2: No

Reviewer #3: No

---

## [Author Response · Author response to Decision Letter 1]

26 Feb 2024

Dear Editor and reviewers,

We appreciate all of the valuable comments from the reviewers of our work. We have revised our manuscript according to the reviewers’ comments, questions, and suggestions. We believe that the manuscript has been further improved.

Attached below are detailed responses to all the reviewer’s comments. The responses are shown in yellow and italicized. Please let us know if you still have any questions or concerns about the manuscript. We will be happy to address them, now promptly

Reviewer #3

Thank you very much for carrying out most of the suggested revisions. There are still a few revisions that need to be addressed to further improve the quality of the manuscript.

Comment 1. My “ introduction” was meant to be a preamble to my review and not meant to be incorporated into the abstract. Hence it was not part of the suggested revisions. However, if the authors find it useful to include it in the introduction of the abstract, that is fine with me. While the authors have included the statements (from my introduction) in the abstract within the main manuscript, the statements not in the abstract that is in the preliminary pages of the manuscript.

Response: Thank you to the reviewer for the comment, we found the preamble interesting and applicable to our introduction and thought we could use it. It is appreciated that using these statements is fine by the reviewer. Also, the statement has been included and referenced in the introduction section as recommended by the reviewer. Kindly see lines 75 – 76 “Use of safe and effective delivery services including place of delivery is an important component of the Safe Motherhood concept (1)”

Comment 2. In response to suggested minor revision # 7 the authors wrote: Kindly see lines 297 – 298 “ Overall, 340 women were surveyed in the study and 91.2% (310/340) agreed to participate in the study”.

The authors should kindly state at least some of the reasons why people were excluded.

Response: The reason for which some women were excluded from the study has been stated under the study population and eligibility section of the methods. Kindly see lines 116 – 117 “Also, women who met our inclusion criteria but refused to participate for personal or health reasons were excluded from the study”

Comment 3. In response to suggested minor revision #10, the authors wrote: “The variable age was collected as a categorical variable, this makes it impossible to present the summary statistics as proposed by the reviewer.”

It is important to state the mean and standard deviation for age as a continuous variable. Age as a continuous variable should not have been categorized/grouped at the data collection stage. The authors should have collected individual ages and grouping/categorization done at the analysis stage. If this was not done, there are still ways of estimating the mean and standard deviation of the data such as using the midpoint of each age group and the respective frequencies. The authors should kindly consult a statistician on how to do this.

Response: The research team engaged the services of a statistician and has added the mean and standard deviation of age as suggested by the reviewer. Kindly see line 194 and table 2 for the revision “The average age of the women was 30.9 ± 6.2 years”

Comment 4. In response to suggested major revision #1, the authors stated that:

The sample size was based on the number of deliveries recorded since our target population was women who had given birth a year prior to the study. So, although there is a large population of WRA in the region, the estimated number of deliveries in the region is much lower. Also, the design effect was adjusted for in the data analysis, where robust standard errors using the type of community as a clustering variable was conducted.

I think my fundamental point is that a sample estimation for predictors of institutional deliveries that makes use of only the prevalence of institutional deliveries without taking into account any of the predictors of institutional deliveries is problematic and may not be adequately powered to detect these predictors. The use of robust standard errors will not adequately deal with the issue of community-based sampling. Given that the data has already been collected, one way round this is to discuss the inadequately powered sample size as a limitation of the study.

Response: We have added to our discussion a limitation of inadequately powered sample size in our study as recommended by the reviewer. Kindly see lines 312 – 314 “Also, the use of only the prevalence of institutional deliveries estimates in calculating the study sample size without taking into account any of the predictors of institutional deliveries introduced the limitation of inadequate power in our study”

Comment 5. In response to suggested major revision #2, the authors stated that:

The strategy used in selecting variables for the adjusted logistic regression analysis has been revised. Kindly see lines 270 – 271 “The adjusted logistic regression model's variables were selected using the stepwise regression approach”.

Which technique was used in adding or eliminating the variables in the stepwise approach? In the two common variable selection approaches in logistic regression (forward selection and backward elimination (selection) methods), the final multivariable model includes variables that are significantly associated with the outcome of interest (based on the authors’ criteria, here p<0.05, using a model fitness test). This will typically exclude all variables that were not significant in the univariable analysis (unless they were considered to be a priori). In addition, it may exclude some variables which were significant in the univariable analysis but were not significantly associated with outcome of interest in the multivariable model based on the model fitness test. Hence, the following variables which were not significant in the univariable model (all p>0.05) should NOT be included in the final multivariable model: age group, religion, educational level, occupation, husband occupation, National Health Insurance Scheme ownership, and attitudes of husbands towards facility delivery. The authors can please refer to:

Hosmer Jr DW, Lemeshow S and Sturdivant RX (2013). Applied Logistic Regression. 3rd Edition. New York: John Wiley and Sons, Inc.

Response: A forward stepwise variable selection approach was used in selecting variables for the multivariate logistic regression. Variables such as age group, religion, educational level, occupation, husband occupation, National Health Insurance Scheme ownership, and attitudes of husbands towards facility delivery which were not significant at the univariate level have been excluded from the final multivariate logistic regression model as suggested by the reviewer. Kindly see lines 223 – 228 and Table 4 “At the multivariate logistic regression analysis level, being married (aOR = 5.54, 95%CI: 3.03 - 10.14), the presence of skilled health personnel (aOR = 2.65, 95%CI: 1.42 - 4.94), the positive attitude of health workers towards their clients (aOR = 1.96, 95%CI: 1.08 - 3.54) and the positive community perception of health facility delivery (aOR = 3.17, 95%CI: 1.34 - 7.47) were associated with increased odds of delivering in a health facility.” Also, the use of the forward stepwise approach in selecting the variables has been indicated in the methods section. Kindly see lines 178 – 179 “The adjusted logistic regression model's variables were selected using forward stepwise variable selection approach”

---

## [Decision Letter · Decision Letter 2]

21 Aug 2024

PONE-D-23-14896R2Predictors of Institutional Delivery Service Utilization among Women in Northern Region of GhanaPLOS ONE

Dear Dr. Mohammed,

Thank you for submitting your manuscript to PLOS ONE. After careful consideration, we feel that it has merit but does not fully meet PLOS ONE’s publication criteria as it currently stands. Therefore, we invite you to submit a revised version of the manuscript that addresses the points raised during the review process.

**ACADEMIC EDITOR: ** Authors should kindly address the following concerns

**Introduction **

The authors cited the 2019 facility delivery rate in Ghana. Can they look at the current rate from the Ghana Demographic Health Survey and argue from there very well?Can they also state the regional institutional delivery rate in their argument, especially in the problem statementThe study gap is not very clear. The authors should clearly state some of the articles in the area, especially in Ghana and the Northern part of the country, and their gap.The authors should indicate the rationale of the study. That piece is missing in the introduction.There should be a theory for the study.

**Methodology**

I believe some districts may have more communities than others, so selecting 16 communities from each district would bias the sampling; how did the authors deal with districts with more communities?  Can the authors state the lowest and highest number of communities for the districts selected?The authors indicated that the kth interval used in selecting households can be specific on the number.This statement, “*For each community, the sampling interval (K) was determined as the total number of houses divided by the number of women to be sampled from the community,* ” and another statement, “*A respondent was recruited at the starting point, and every fifth house starting from the southward direction was visited* ” are conflicting statements. This statement needs to be well explained. Does this mean that the sampling interval was used differently in each community? The authors should add a supplementary sheet on how the sampling was done concerning the communities and the selection of houses.There should be a section on the measurement of variables. Clearly describe how the dependent variable and the independent variables were measured.Authors should state when the data was collected (month and year).

**Results**

The authors should elaborate clearly on how the variables were selected for the logistic regression (Table 4). What were the criteria for reducing the variables to only 4?

**Discussion **

Link the theory to the discussion.

**Recommendation **

“The recommendation of the study is not very strong. For instance, *“To increase health facility delivery, we recommend the Ministry of Health should institute a policy reform with a well-defined care package targeting unmarried pregnant women and health workers with a negative attitude and community perception.” * Can authors suggest a policy in that direction? They should add other recommendations, especially given the context that most of the communities are rural

We look forward to receiving your revised manuscript.

Kind regards,

Martin Wiredu Agyekum, PhD

Guest Editor

PLOS ONE

Journal Requirements:

Reviewers' comments:

Reviewer's Responses to Questions

**Comments to the Author**

1. If the authors have adequately addressed your comments raised in a previous round of review and you feel that this manuscript is now acceptable for publication, you may indicate that here to bypass the “Comments to the Author” section, enter your conflict of interest statement in the “Confidential to Editor” section, and submit your "Accept" recommendation.

Reviewer #1: All comments have been addressed

Reviewer #3: All comments have been addressed

2. Is the manuscript technically sound, and do the data support the conclusions?

Reviewer #1: Yes

Reviewer #3: Yes

3. Has the statistical analysis been performed appropriately and rigorously? 

Reviewer #1: Yes

Reviewer #3: Yes

4. Have the authors made all data underlying the findings in their manuscript fully available?

Reviewer #1: Yes

Reviewer #3: Yes

5. Is the manuscript presented in an intelligible fashion and written in standard English?

Reviewer #1: Yes

Reviewer #3: Yes

6. Review Comments to the Author

Reviewer #1: Your manuscript has been appropriately revised and reviewer's comments has been .adequately addressed.

Reviewer #3: No further comments .

7. PLOS authors have the option to publish the peer review history of their article (what does this mean? ). If published, this will include your full peer review and any attached files.

**Do you want your identity to be public for this peer review?** For information about this choice, including consent withdrawal, please see our Privacy Policy .

Reviewer #1: **Yes: ** Mumuni Kareem

Reviewer #3: No

---

## [Author Response · Author response to Decision Letter 2]

20 Oct 2024

Dear Editor and reviewers,

We appreciate all of the valuable comments from the reviewers of our work. We have revised our manuscript according to the reviewers’ comments, questions, and suggestions. We believe that the manuscript has been further improved.

Attached below are detailed responses to all the reviewer’s comments. The responses are shown in green and italicized. Please let us know if you still have any questions or concerns about the manuscript. We will be happy to address them, now promptly

Introduction

Comment: The authors cited the 2019 facility delivery rate in Ghana. Can they look at the current rate from the Ghana Demographic Health Survey and argue from there very well?

Response: Data from the 2022 Demographic Health Survey has been used as recommended “

According to the 2022, demographic health survey in Ghana, 86% of live births in the 2 years preceding the survey delivered in a health facility. This implies more than 10% of children are still delivered outside the health facility setting” kindly see page 3, lines 61 - 66

Comment: Can they also state the regional institutional delivery rate in their argument, especially in the problem statement

Response: The regional institutional delivery rate has been added the problem statement as suggested “The Northern region of Ghana continues to experience one of the lowest institutional delivery rates in the country, with only 70.3% of births occurring in health facilities (DHS 2022). Surveys conducted in Brong Ahafo region and Chereponi district of northern Ghana, among 138 and 440 women respectively, revealed health facility deliveries ranged from 38 – 52% (4,5).” Kindly see page 3, lines 65 - 68

Comment: The study gap is not very clear. The authors should clearly state some of the articles in the area, especially in Ghana and the Northern part of the country, and their gap.

Response: The articles and their gaps have been included as suggested “Also, published studies from the region on institutional delivery service utilisation are mostly descriptive studies or focus on particular districts (12,13) and do not thoroughly investigate the correlates of institutional delivery service utilisation at the regional level” kindly see page 5, lines 93 - 96

Comment: The authors should indicate the rationale of the study. That piece is missing in the introduction.

Response: The rationale for the study has been included in the introduction “Despite the established consequences of deliveries outside a health facility among WRA, there is a dearth of knowledge on the prevalence and predictors of health facility delivery in the Northern region. Also, published studies from the region on institutional delivery service utilisation are mostly descriptive studies or focus on particular districts (12,13) and do not thoroughly investigate the correlates of institutional delivery service utilisation at the regional level. Understanding the factors influencing health facility deliveries in the Northern region is crucial for designing targeted interventions to improve maternal health outcomes. This study aims to identify the prevalence and predictors of institutional delivery service utilization at the regional level, providing evidence-based insights that can guide policymakers in formulating strategies to increase health facility deliveries and ultimately reduce maternal and neonatal mortality in the region.” Kindly see page 5, lines 90 - 100

Comment: There should be a theory for the study.

Response: The theoretical framework of the study has been included “The theoretical framework of the study will be guided by the Health Belief Model (HBM) and elements from Andersen's Behavioral Model of Health Services Use. These models offer a combined approach to understanding how individual beliefs, social context, and access to resources influence institutional delivery. This framework will guide the study in identifying key predictors of service utilization and provide a basis for designing targeted interventions that address specific barriers and motivators identified in the Northern region of Ghana.” Kindly see page 5, lines 103 - 106

Methodology

Comment: I believe some districts may have more communities than others, so selecting 16 communities from each district would bias the sampling; how did the authors deal with districts with more communities?

Response: A stratified sampling approach grouping the communities into urban and rural areas was used to enable us to understand the problem in both urban and rural areas “The communities were categorized under stratum A (list of urban communities in the district) and stratum B (list of rural communities in the district). In each of the districts, the names of communities in each stratum were written on pieces of paper and placed in a box, shaken and one piece of paper randomly selected from each stratum.” Kindly see page 7, lines 156 - 159

Comment: Can the authors state the lowest and highest number of communities for the districts selected?

Response: The number of communities in the districts selected has been stated as suggested by the reviewer “Communities in the selected districts ranged from 20 – 35. The communities were categorized under stratum A (list of urban communities in the district) and stratum B (list of rural communities in the district). In each of the districts, the names of communities in each stratum were written on pieces of paper and placed in a box, shaken and one piece of paper randomly selected from each stratum.” Kindly see page 7, lines 156 - 159

Comment: The authors indicated that the kth interval used in selecting households can be specific on the number.

Response: The kth interval varied based on the number of households in each of the communities. We calculated the kth interval for each community

Comment: This statement, “For each community, the sampling interval (K) was determined as the total number of houses divided by the number of women to be sampled from the community,” and another statement, “A respondent was recruited at the starting point, and every fifth house starting from the southward direction was visited” are conflicting statements. This statement needs to be well explained. Does this mean that the sampling interval was used differently in each community? The authors should add a supplementary sheet on how the sampling was done concerning the communities and the selection of houses.

Response: The kth interval varied based on the number of households in each of the communities. We calculated kth interval for each community. Also the statement on the kth household and the fifth household has been revised. Kindly see page 8, lines 170 - 180

Comment: There should be a section on the measurement of variables. Clearly describe how the dependent variable and the independent variables were measured.

Response: “Dependent variable: The dependent variable in the study was the place of delivery in the most recent birth. The variable was binary (Health facility/Home).

Independent variables: The independent variables were divided into three groups, sociodemographic characteristics of the participants, health facility-level factors, and community-level factors. The sociodemographic characteristics included occupation, religion, marital status, educational level, income level, husband education, and husband occupation. Health facility-level factors include the availability of a health facility, the attitude of health workers, the presence of health workers at post, and the distance to the nearest health facility. The community-level factors included the availability of TBAs in the community, community perception, the attitude of partners towards health facility delivery, and availability of transportation to health facilities.” Kindly see page 7, lines 135 - 144

Comment: Authors should state when the data was collected (month and year).

Response: The data collection period, month and year have been added “The data collection period was March 10th – April 2nd 2023”. Kindly see page 6, line 115

Results

Comment: The authors should elaborate clearly on how the variables were selected for the logistic regression (Table 4). What were the criteria for reducing the variables to only 4?

Response: The adjusted logistic regression model's variables were selected using a forward stepwise variable selection approach. Only variables which were significant at the crude level were presented in the table

Discussion

Comment: Link the theory to the discussion.

Response: The discussions have been linked to the Health Belief Model (HBM) and elements from Andersen's Behavioral Model of Health Services Use. Kindly see page 16

Recommendation

Comment: “The recommendation of the study is not very strong. For instance, “To increase health facility delivery, we recommend the Ministry of Health should institute a policy reform with a well-defined care package targeting unmarried pregnant women and health workers with a negative attitude and community perception.” Can authors suggest a policy in that direction? They should add other recommendations, especially given the context that most of the communities are rural

Response: More recommendations towards improving facility delivery, particularly in the rural communities have been added “Also, the government should provide incentives for health workers working in rural communities.” Kindly see page 20, lines 353 - 354

---

## [Editor Report · Decision Letter 3]

24 Apr 2025

Predictors of Institutional Delivery Service Utilization among Women in Northern Region of Ghana

PONE-D-23-14896R3

Dear Dr. Mohammed,

We’re pleased to inform you that your manuscript has been judged scientifically suitable for publication and will be formally accepted for publication once it meets all outstanding technical requirements.

Kind regards,

Mubarick Nungbaso Asumah, MPhil, Bsc

Academic Editor

PLOS ONE
---

## [Editor Report · Acceptance letter]

PONE-D-23-14896R3

PLOS ONE

Dear Dr. Mohammed,

I'm pleased to inform you that your manuscript has been deemed suitable for publication in PLOS ONE. Congratulations! Your manuscript is now being handed over to our production team.

Kind regards,

on behalf of

Dr. Mubarick Nungbaso Asumah

Academic Editor

PLOS ONE